# DOTS: LEARNING TO REASON DYNAMICALLY IN LLMS VIA OPTIMAL REASONING TRAJECTORIES SEARCH

**Murong Yue**$^{\alpha *}$ **Wenlin Yao**$^{\beta}$ **Haitao Mi**$^{\beta}$ **Dian Yu**$^{\beta}$ **Ziyu Yao**$^{\alpha}$ **Dong Yu**$^{\beta}$

$^{\alpha}$George Mason University
$^{\beta}$Tencent AI Lab, Bellevue
`{myue,ziyuyao}@gmu.edu`
`{wenlinyao,haitaomi,yudian,dyu}@global.tencent.com`

## ABSTRACT

Enhancing the capability of large language models (LLMs) in reasoning has gained significant attention in recent years. Previous studies have demonstrated the effectiveness of various prompting strategies in aiding LLMs in reasoning (called "reasoning actions"), such as step-by-step thinking, reflecting before answering, solving with programs, and their combinations. However, these approaches often applied static, predefined reasoning actions uniformly to all questions, without considering the specific characteristics of each question or the capability of the task-solving LLM. In this paper, we propose **DOTS**, an approach enabling LLMs to reason Dynamically via Optimal reasoning Trajectories Search, tailored to the specific characteristics of each question and the inherent capability of the task-solving LLM. Our approach involves three key steps: i) defining atomic reasoning action modules that can be composed into various reasoning action trajectories; ii) searching for the optimal action trajectory for each training question through iterative exploration and evaluation for the specific task-solving LLM; and iii) using the collected optimal trajectories to train an LLM to plan for the reasoning trajectories of unseen questions. In particular, we propose two learning paradigms, i.e., fine-tuning an external LLM as a planner to guide the task-solving LLM, or directly fine-tuning the task-solving LLM with an internalized capability for reasoning actions planning. Our experiments across eight reasoning tasks show that our method consistently outperforms static reasoning techniques and the vanilla instruction tuning approach. Further analysis reveals that our method enables LLMs to adjust their computation based on problem complexity, allocating deeper thinking and reasoning to harder problems. Our code is available at `https://github.com/MurongYue/DOTS`.

## 1 INTRODUCTION

Large Language Models (LLMs) have demonstrated remarkable performance in solving complex reasoning tasks (Rae et al., 2021; Lewkowycz et al., 2022; Zhong et al., 2023), such as math reasoning (Imani et al., 2023; Ahn et al., 2024), symbolic reasoning (Kojima et al., 2022), and commonsense reasoning (Krause & Stolzenburg, 2023; Zhao et al., 2024). The dominant approaches to eliciting reasoning capability in LLMs mainly fall into two categories, i.e., instruction tuning and prompt engineering. Instruction tuning (Wang et al., 2022) collects question-answer pairs about the reasoning task and employs supervised fine-tuning to optimize an LLM for better reasoning performance (Yue et al., 2024; Tang et al., 2024), with recent effort focusing on improving the scale and the quality of the fine-tuning data (Luo et al., 2023; Peng et al., 2023; Yue et al., 2023; 2024; Chan et al., 2024). Prompt engineering instead aims to design better prompts to elicit the reasoning capability of an LLM without updating its parameters. The Chain-of-Thought (CoT) approach (Wei et al., 2022; Kojima et al., 2022) prompts an LLM to answer the reasoning question step by step in natural language, and program-aided approaches (Chen et al., 2022; Gao et al., 2023) prompt the LLM to write executable code and leverage an interpreter to execute code for obtaining the final result. Besides,

---

$^{*}$Work done during Murong Yue's internship at Tencent AI Lab.

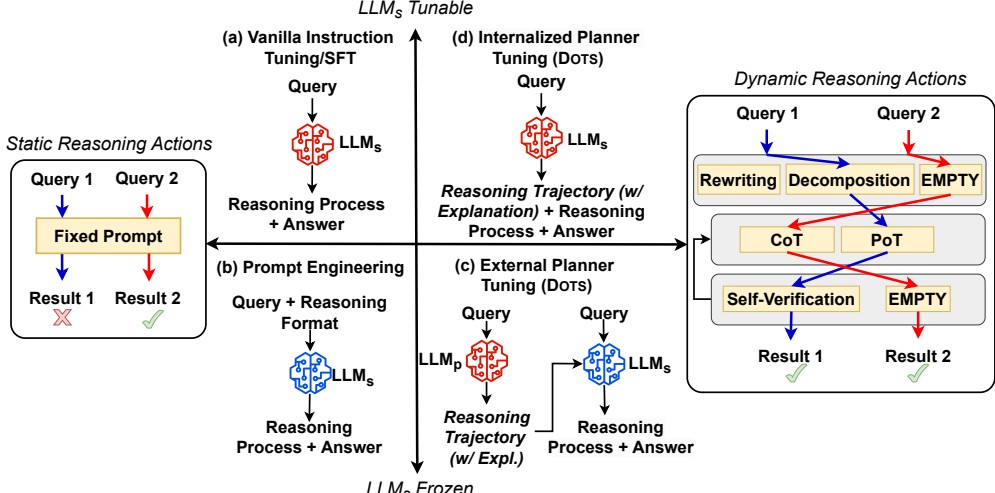

Figure 1: A comparison of different paradigms of LLM reasoning. Unlike prior approaches with predefined, static reasoning actions, DOTS dynamically plans for the optimal reasoning trajectory per each question and the specific task-solving LLM ($LLM_s$). In particular, DOTS encompasses two inference setups, i.e., external planner tuning (c) and internalized planner tuning (d), depending on whether to introduce an external LLM as a planner ($LLM_p$) or to internalize the trajectory planning capability into the same solver LLM ($LLM_s$). (🧩: tunable; 🧩: frozen)

prompting the LLM to decompose the question before answering it (Radhakrishnan et al., 2023; Zhou et al., 2023), or to verify the solution before returning it as the final answer (Madaan et al., 2024), has also been proven effective in specific reasoning tasks.

However, both types of approaches suffer from a critical limitation, i.e., being unable to *dynamically decide the best reasoning strategies*.

Instruction-tuned LLMs are constrained by the reasoning formats in their training data (e.g., CoT (Luo et al., 2023)), as illustrated by a GPT-4o case in Appendix B. FireAct (Chen et al., 2023a) shows that fine-tuned LLMs can switch between reasoning strategies like CoT and ReAct, but our experiments reveal that training with varied trajectories alone is insufficient for optimal results.

On the other hand, current prompt engineering approaches assume predefined prompting strategies and uniformly apply the same to every question. However, different types of questions are better suited to different reasoning strategies (Zhao et al., 2023), and the effectiveness of a prompting approach also depends on the inherent capability of the task-solving LLM (e.g., LLMs pre-trained on code data are better at programming-aided reasoning). Consequently, the same prompt may not be equally effective for every question and every LLM.

In this paper, we present DOTS, an approach empowering LLMs to actively select optimal reasoning actions for given questions and the task-solving LLM (Figure 1). We begin by constructing atomic reasoning action modules, which are composed to generate multiple potential reasoning action trajectories. Then we collect the training data by searching for an optimal (in terms of both its success rate and the number of reasoning actions needed) action trajectory through numerous explorations and evaluations. This optimal trajectory is tailored to the specific task-solving LLM. Subsequently, we employ supervised fine-tuning to train an LLM in determining the optimal reasoning action trajectory. We implement this approach in two distinct setups: (1) For closed-source or computationally costly task-solving LLMs, we fine-tune a smaller LLM as an external planner to predict optimal reasoning actions for the task-solving LLM; (2) For open-source and small-size LLMs, we fine-tune the task-solving LLM itself to plan on the reasoning actions to take before solving the reasoning task, internalizing the autonomous planning capability directly into the LLM. This dual approach allows for flexible application across different LLM accessibility constraints.

Our experimental results demonstrate the efficacy of our proposed method in enhancing the reasoning capabilities of LLMs. We conducted extensive evaluations across multiple LLMs (GPT-4o-mini, Llama3-70B-Instruct, and Llama3-8B-instruct (Dubey et al., 2024)) and a diverse set of reasoning

tasks, encompassing in-distribution, few-shot, and out-of-distribution scenarios. The results reveal that DOTS consistently outperforms static prompt engineering techniques and vanilla instruction tuning methods across various reasoning challenges. Through a comprehensive ablation study, we validate the significance of each component in our methodology. Moreover, our analysis of reasoning action distributions highlights that our method can adapt to the specific characteristics of reasoning questions and the inherent capability of task-solving LLMs. We further confirm that our method incurs minimal additional financial costs. Lastly, we showcase that LLMs can naturally develop the capacity to allocate more computational resources to complex problems through a process of exploration and learning, without explicit guidance.

## 2 DOTS: LEARNING TO REASON DYNAMICALLY

### 2.1 OVERVIEW

Our goal is to enable LLMs to select the most effective reasoning actions autonomously. Denote $LLM_s$ as the task-solving LLM, $Q$ as the input query, $p$ as the reasoning action trajectory path, $E$ as the explanation for a trajectory, and $R$ as the reasoning process leading to the final answer $y$. Our approach encompasses two setups during the inference stage (Figure 1):

**External Planner Tuning**  This setup is designed for scenarios where the solver ($LLM_s$) is a closed-source LLM or is computationally costly to train. As depicted in Figure 1 (c), we train an external planner, denoted as $LLM_p$, to determine the optimal reasoning actions:

$$(E, p) = LLM_p(Q; \theta_p) \qquad (1)$$

where $\theta_p$ is the parameters of $LLM_p$. We empirically found that training the planner to explain its trajectory selection ($E$) helps its learning. Upon obtaining reasoning actions, the solver $LLM_s$ parameterized by $\theta_s$ then proceeds to generate the reasoning process $R$ and the final answer $y$:

$$(R, y) = LLM_s(Q, p; \theta_s) \qquad (2)$$

**Internalized Planner Tuning**  This setup is designed for task-solving LLMs ($LLM_s$) that are open-source and small-size. In this case, we propose to *internalize* the trajectory planning capability into the task-solving LLM by training it to simultaneously learn to plan and learn to perform the reasoning task. As shown in Figure 1 (d), the final answer $y$ is obtained by:

$$(E, p, R, y) = LLM_s(Q; \theta_s) \qquad (3)$$

An overview of DOTS's learning process is presented in Figure 2, consisting of three key steps: (i) **Defining atomic reasoning modules:** We define several atomic reasoning modules, each representing a distinct reasoning action, (ii) **Searching for optimal action trajectories:** We conduct explorations and evaluation of various reasoning paths to identify optimal reasoning actions for questions in the training data, and (iii) **Fine-tuning LLMs to plan for optimal reasoning trajectories:** We fine-tune LLMs to autonomously plan the reasoning action trajectory under the two aforementioned setups. In what follows, we elaborate on each step.

Table 1: Prompt engineering methods with different reasoning actions. Our method could dynamically select reasoning actions among all of them.

| Prompting Method | Analysis Layer | | Solution Layer | | Verification Layer |
|---|---|---|---|---|---|
| | Rewriting | Decomposition | NL | Program | Verification |
| CoT (Wei et al., 2022) | ✗ | ✗ | ✓ | ✗ | ✗ |
| PoT (Chen et al., 2022) | ✗ | ✗ | ✗ | ✓ | ✗ |
| LTM (Zhou et al., 2023) | ✗ | ✓ | ✓ | ✗ | ✗ |
| R&R (Deng et al., 2023) | ✓ | ✗ | ✓ | ✗ | ✗ |
| Self-Refine (Madaan et al., 2024) | ✗ | ✗ | ✓ | ✓ | ✓ |
| Self-Verification (Weng et al., 2022) | ✗ | ✗ | ✓ | ✗ | ✓ |
| PromptAgent (Wang et al., 2023) | ✓ | ✓ | ✓ | ✗ | ✗ |
| DOTS (**ours**) | ✓ | ✓ | ✓ | ✓ | ✓ |

### 2.2 DEFINING ATOMIC REASONING ACTIONS MODULES

Prior studies have validated the effectiveness of various reasoning strategies (Table 1). We build on top of them and categorize the existing strategies as reasoning actions across three layers:

**Analysis Layer**  Actions in this layer enable the LLM to analyze the input query before attempting to solve it, including (1) `Query rewriting`: reformulating the query to enhance comprehension, and (2) `Query decomposition`: breaking down the initial question into multiple, more manageable sub-questions. We denote the action taken in this layer as $A_a$.

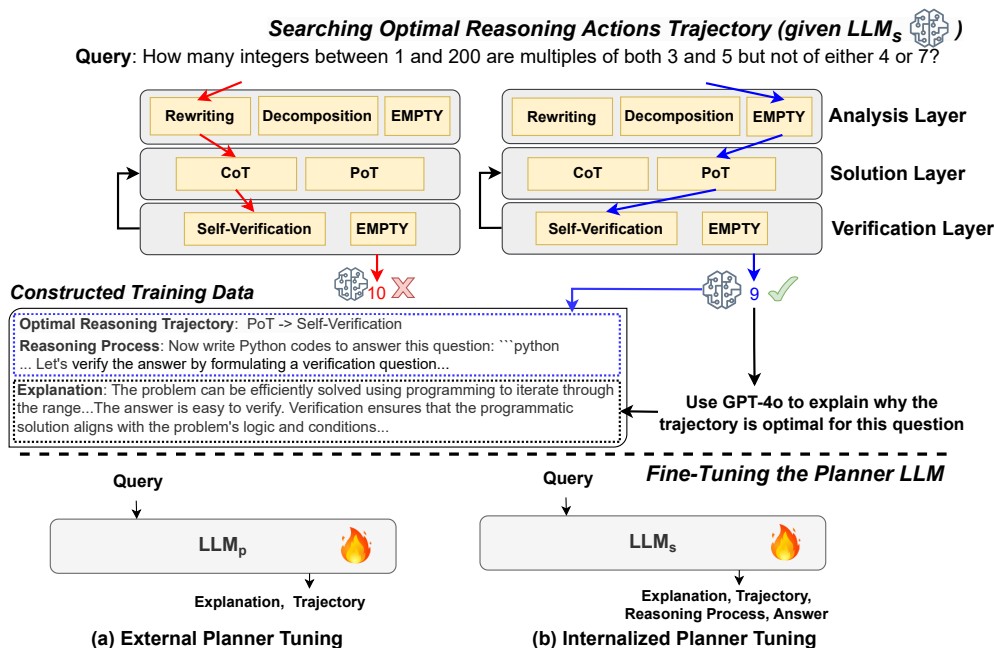

Figure 2: The training process of DOTS, including searching for the optimal reasoning trajectories for questions in the training set and fine-tuning the internalized/external planner LLM.

**Solution Layer**  Actions in this layer consider variants in the reasoning format. Prior works showed that different queries are better solved following different reasoning processes (Zhao et al., 2023). In our work, we consider the most commonly adopted formats, i.e., (1) CoT: solving the question step-by-step in natural language, and (2) PoT: addressing the question through code generation. We denote the action chosen in this layer as $A_t$.

**Verification Layer**  Finally, the verification layer is responsible for checking the correctness of the proposed solution. It is particularly useful for problems where verification is significantly easier than solving the problem itself, e.g., the Game of 24 (Yao et al., 2024). Therefore, we set a Self-Verification action module in this layer. If this module determines that the reasoning process from the solution layer is incorrect, the LLM will revert to the solution layer to reattempt to solve the problem. During this reattempt, the LLM is provided with both the initial answer and the feedback from the verifier explaining why the initial answer was incorrect. The process continues until the verifier confirms that the answer is correct or the pre-defined maximum number of iterations for self-verification is reached. We denote the action taken in this layer as $A_v$.

We observe that introducing excessive reasoning actions can lead to increased latency, and even sometimes result in incorrect answers. To mitigate this issue, we introduce an Empty action in both the analysis and the verification layers, allowing the LLM to bypass these two layers when dealing with simple questions. Detailed prompts for each module are provided in Appendix D.

## 2.3 SEARCHING FOR OPTIMAL REASONING ACTION TRAJECTORIES

To teach the external/internalized planner to plan for the optimal reasoning trajectory, we start by constructing training data containing questions and their optimal action trajectories for the specific task-solving LLM. We obtain this by iteratively searching all possible reasoning trajectories for each question, including exploring the current paths and pruning paths that are unlikely to be optimal. The task-solving LLM is used during this search process to generate answers to make the reasoning trajectory align with their intrinsic ability to perform different reasoning actions effectively.

This searching process is shown in Algorithm 1. Given the query and ground-truth answer sourced from the training data, the process runs iteratively. In each iteration, the algorithm considers either the full set of candidate trajectories (for iteration $k = 1$) or the current best subset (for iteration $k > 1$). Each candidate trajectory is executed for $N_{eval}$ times with a non-zero temperature to obtain a more reliable evaluation of its success rate. We then sort the current subset of trajectories by its

---

**Algorithm 1** Searching for the Optimal Reasoning Action Trajectory

---

**Require:** Input query $Q$ and ground true answer $y^*$, solver $LLM_s$, max iteration $K$, number of evaluations $N_{eval}$, and number of candidate trajectories to retain in each iteration $N_1, N_2, \cdots, N_K$.
**Ensure:** Optimal action trajectory path $p^*$ for query $Q$ and solver $LLM_s$.
 1: Initialized candidate trajectory set $\mathcal{P} \leftarrow \{(A_a = \text{Empty}, A_t = \text{CoT}, A_v = \text{Empty}), \cdots\}$;
 2: Initialize the record of the accumulated success rate of each candidate trajectory: $\mathcal{R} \leftarrow \{p : 0 \mid p \in \mathcal{P}\}$;
 3: **for** iteration $k = 1$ to $K$ **do**
 4:     **for all** $p \in \mathcal{P}$ **do**
 5:         Execute the trajectory $p$ against $Q$ and $LLM_s$ for $N_{eval}$ times with non-zero temperature and obtain an average success rate $r_p$ (compared to the true answer $y^*$);
 6:         Update the accumulated success rate of $p$: $\mathcal{R}[p] \leftarrow \frac{\mathcal{R}[p] \cdot (k-1) \cdot N_{eval} + r_p \cdot N_{eval}}{N_{eval} \cdot k}$;
 7:     **end for**
 8:     Sort $\mathcal{P}$ first by the accumulated success rate $\mathcal{R}[p]$ and then the trajectory length $|p|$ in ascending order;
 9:     Reset $\mathcal{P} \leftarrow$ top $N_k$ trajectories in $\mathcal{P}$.
10: **end for**
11: **Return** $p^* \leftarrow \arg\max_{p \in \mathcal{P}} \mathcal{R}[p]$.

---

success rate accumulated from the past $k$ iterations and then the trajectory length to encourage a shorter trajectory (which is thus computationally more efficient). Only the top $N_k$ candidates will be retained and rolled over to the next iteration of the assessment. In practice, we opt for a smaller $N_{eval}$ and run the search for multiple iterations, as opposed to finishing the search with a larger $N_{eval}$ in one iteration, as the latter incurs a much larger cost ($N_{eval} \times |\mathcal{P}_0|$ with a large $N_{eval}$ vs. $N_{eval} \times (|\mathcal{P}_0| + N_1 + \cdots + N_{K-1})$ with a small $N_{eval}$ in our algorithm).

In the process of validating various trajectories for each question, we exclude instances where *any* trajectory solves the query or *all* fail to do so, as they do not contribute to the planner LLM's trajectory planning learning. After identifying the best reasoning trajectory, we leverage GPT-4o to verbally explain why the trajectory is optimal. Our prompt is shown in Appendix D. This process is applied to all instances in the training data, giving us tuples of query $Q$, ground true answer $y^*$, optimal trajectory $p^*$, and its explanation $E$. For internalized planner tuning, we collect the reasoning process $R$ when running the solver $LLM_s$ following the optimal trajectory $p^*$.

### 2.4 Learning to Plan for Optimal Reasoning Trajectories

Having obtained the optimal trajectories, we then use supervised fine-tuning with cross-entropy loss to train the planner LLM to predict optimal trajectories for input questions and the specific solver LLM. For external planner tuning, a lightweight $LLM_p$ is trained to predict a concatenation of the explanation and the optimal trajectory (Eq 1); for internalized planner tuning, the solver $LLM_s$ is trained to predict the explanation, the optimal trajectory, the reasoning process collected from $LLM_s$ itself, and the true answer $y^*$ (Eq 3).

## 3 Experiment

### 3.1 Experimental Setup

**Datasets**  We evaluate the effectiveness of our method across multiple datasets and various reasoning tasks. Based on the distribution of the training and testing data, we divide the evaluation into three settings as shown in Table 2: *In-distribution setting* evaluates the model that resembles what it has seen during training. *Few-shot setting* aims to evaluate whether our proposed method can effectively learn from a small amount of labeled data. In the real

Table 2: Overview of our evaluation datasets.

| Dataset | Distribution and Size | Task Type |
|---|---|---|
| MATH | In Distribution, Many shot | math |
| BBH | | mixture |
| Game of 24 | In Distribution, Few-shot | numerical |
| TheoremQA | | scientific |
| Deepmind Math | | math |
| MMLU-pro | | scientific |
| StrategyQA | Out-of-Distribution | common sense |
| DROP | | multi-hop |

world, it is often difficult to obtain large amounts of in-domain training data across different tasks, but a small number of cases can be annotated. *Out-of-distribution (OOD) setting* further evaluates whether the model can handle scenarios it was not explicitly trained for, testing its ability to generalize beyond the training set. For the training data, we use the MATH (Hendrycks et al., 2021) training set. For the few-shot learning, we select 4 examples from each category of BBH (Suzgun

Table 3: Accuracy (%) of the external planner tuning on in-distribution and few-shot datasets. The reasoning format $\mathcal{L}$ represents language, and $\mathcal{P}$ means program.

| Method | Tuning | Reasoning Format | MATH | BBH | Game of 24 | TheoremQA | Average |
|---|---|---|---|---|---|---|---|
| **External Planner: Llama-3-8B-Instruct; Solver: Llama-3-70B-Instruct** | | | | | | | |
| CoT | ✗ | $\mathcal{L}$ | 50.4 | 72.7 | 27.5 | 27.4 | 44.5 |
| LTM | ✗ | $\mathcal{L}$ | 50.1 | 73.8 | 24.9 | 28.8 | 44.4 |
| PA | ✓ | $\mathcal{L}$ | 52.5 | 72.9 | 26.8 | 28.8 | 45.3 |
| PoT | ✗ | $\mathcal{P}$ | 54.7 | 65.8 | 63.9 | 31.1 | 53.9 |
| Self-refine | ✗ | $\mathcal{L}, \mathcal{P}$ | 55.9 | 71.4 | **68.3** | 30.8 | 56.6 |
| **DOTS: External** | ✓ | $\mathcal{L}, \mathcal{P}$ | **57.7** | **77.3** | 67.7 | **31.2** | **58.5** |
| **External Planner: Llama-3-8B-Instruct; Solver: GPT4o-mini** | | | | | | | |
| CoT | ✗ | $\mathcal{L}$ | 70.2 | 80.3 | 27.7 | 38.9 | 54.2 |
| LTM | ✗ | $\mathcal{L}$ | 72.2 | 79.4 | 25.5 | 36.4 | 53.3 |
| PA | ✓ | $\mathcal{L}$ | 73.5 | 81.1 | 26.7 | 38.9 | 55.1 |
| PoT | ✗ | $\mathcal{P}$ | 67.2 | 73.9 | 61.4 | 35.8 | 59.6 |
| Self-refine | ✗ | $\mathcal{L}, \mathcal{P}$ | 73.7 | 74.8 | **68.7** | 34.6 | 63.0 |
| **DOTS: External** | ✓ | $\mathcal{L}, \mathcal{P}$ | **75.4** | **84.2** | 65.2 | **41.4** | **66.5** |

et al., 2022) as it is composed of 27 diverse tasks,[1] resulting in 108 examples in total, 4 examples from Game of 24 (Yao et al., 2024), and 4 examples from TheoremQA (Chen et al., 2023b) datasets. For the test data, we evaluate the model on the test set of the MATH dataset for the in-distribution setting and on the test sets or hold-out sets of BBH, Game of 24, and TheoremQA for the few-shot learning setting. For the OOD evaluation, we test each approach's generalization ability on Deep-mind Math (Saxton et al., 2019), MMLU-pro (Wang et al., 2024), strategyQA (Geva et al., 2021), and DROP (Dua et al., 2019). All evaluations (unless specified) were conducted when prompting the solver LLMs in zero shot. For answer evaluation, we use the simple-eval[2] for MATH, a standard evaluation for Game of 24 (Yao et al., 2024), and exact string matching for the others.

**Training Setup** For external planner tuning, we utilize Llama-3-8B-Instruct as our planner and GPT-4o-mini and Llama-70B-Instruct as task-solving LLMs. Experiments of internalized planner tuning were conducted with Llama-3-8B-Instruct. For more details, refer to Appendix A.

### 3.2 BASELINES

We include the following highly related baselines in our experiments. (1) CoT (Wei et al., 2022) prompts an LLM to answer step-by-step; (2) PoT (Chen et al., 2022) prompts an LLM to generate Python code and execute the code to get the final answer; (3) Least-to-most (LTM) (Zhou et al., 2023) prompts an LLM to first decompose the question into multiple sub-questions before solving it; (4) Self-refine (Madaan et al., 2024) prompts an LLM to generate the answer and verify and refine the answer by the LLM itself. Madaan et al. (2024) used PoT in solving math questions, therefore we follow their setting to use PoT in generating the initial answer; (5) PromptAgent (PA) (Wang et al., 2023) searches for a better prompt for the specific task based on its training data; this baseline is implemented with the default hyperparameter setting; and (6) Vanilla Supervised Fine-Tuning (Vanilla SFT) uses GPT-4o to generate the CoT reasoning process for questions in the training datasets and then fine-tune the solver LLM to predict the generated reasoning process and the ground-truth answer; this baseline is fine-tuned using the same hyperparameter setting as our internalized planner tuning. The training data for PA, Vanilla SFT, and DOTS are from the same source.

### 3.3 EXTERNAL PLANNER TUNING RESULTS

Table 3 presents the results of using the external planner, which suggest that:

**External planner tuning outperforms other methods on the in-domain task** Our method achieves 57.7% accuracy with Llama-3-70b-Instruct and 75.4% accuracy with GPT-4o-mini on MATH, achieving significant improvement than baselines. This suggests that DOTS is robust across different LLMs and it can significantly enhance the LLM's zero-shot reasoning ability. The improvement from DOTS remains consistent as the solver LLM's capabilities increase, indicating DOTS has a long-term value even as LLMs continue to improve rapidly.

**The external planner can learn the appropriate action trajectory with only a few training examples.** On the BBH, DOTS achieves improvements of 3.5% and 3.1% over the best static

---

[1] https://huggingface.co/datasets/lukaemon/bbh
[2] https://github.com/openai/simple-evals.

methods when using Llama-3-70B-Instruct and GPT-4o-mini, respectively. In the Game of 24 and TheoremQA, DOTS also shows slight improvements or performs similarly to the best static method. This indicates that even a small number of cases can help the LLM learn the optimal strategy for the given task. Besides, DOTS demonstrates greater stability across various datasets. Our flexible action trajectory selection demonstrates its advantages on datasets requiring diverse reasoning actions, such as BBH as shown in Appendix C. Conversely, the Game of 24 features a uniform question type, where the predefined static method self-refine is sufficient. While the self-refine excels on Game of 24, it significantly lags behind on other datasets. This reflects the external planner's ability to effectively select the appropriate action trajectory, leading to more robust performance even across tasks with varying reasoning demands.

## 3.4 INTERNALIZED PLANNER TUNING RESULTS

Table 4 presents the results of our internalized planner tuning, where we observed:

**Internalized planner tuning demonstrates superior performance** DOTS outperforms existing methods on average, including prompt engineering methods and vanilla SFT. Notably, our approach surpasses self-refine in the Game of 24, a different observation than the experiments with an external planner (Table 3). We attribute this performance boost to our joint optimization of the trajectory planning and problem-solving processes. Unlike external planner tuning which only updates the external planner ($LLM_p$), internalized planner tuning enables the task-solving LLM to simultaneously learn trajectory planning and accurate reasoning process generation. This highlights that the internalized planner tuning effectively further enhances performance.

**Searching for the optimal reasoning action trajectory helps enhance the utilization of training data** Compared to vanilla SFT, our method consistently shows performance improvements across all datasets, notably achieving an 8.7% increase on BBH. This suggests that, instead of training with a question and step-by-step reasoning process pair, our approach of searching for an optimal action trajectory and generating the corresponding reasoning process to construct training data is superior. This finding indicates that our search methodology could effectively enhance the utilization of training data for reasoning tasks without the need for additional human annotations.

Table 4: Internal planner tuning performance on in-distribution and few-shot datasets.

| Method | Tuning | Reasoning format | MATH | BBH | Game of 24 | TheoremQA | Average |
|---|---|---|---|---|---|---|---|
| **Solver: Llama-3-8B-Instruct** | | | | | | | |
| CoT | ✗ | $\mathcal{L}$ | 29.6 | 48.9 | 12.7 | 14.8 | 26.5 |
| LTM | ✗ | $\mathcal{L}$ | 29.5 | 50.3 | 14.4 | 15.2 | 27.4 |
| PA | ✓ | $\mathcal{L}$ | 31.0 | 47.2 | 11.8 | 15.1 | 26.3 |
| PoT | ✗ | $\mathcal{P}$ | 25.3 | 44.6 | 16.8 | **16.7** | 25.9 |
| Self-refine | ✗ | $\mathcal{L}, \mathcal{P}$ | 28.7 | 46.6 | 17.0 | 15.3 | 30.1 |
| Vanilla SFT | ✓ | $\mathcal{L}$ | 33.9 | 61.0 | 18.5 | 14.8 | 33.6 |
| **DOTS: Internalized** | ✓ | $\mathcal{L}, \mathcal{P}$ | **34.4** | **69.7** | **21.9** | 16.1 | **35.5** |

## 3.5 OUT-OF-DISTRIBUTION EXPERIMENTAL RESULTS

**Our method consistently generalizes well across diverse OOD challenges** As shown in Table 5, DOTS maintains high accuracy across different datasets and models. In contrast, static methods often fluctuate significantly in performance. For instance, despite static methods like CoT showing a slight advantage on MMLU-Pro and StrategyQA over DOTS using the Llama-3-70B-Instruct model, they experience a sharp decline on DeepMind Math. This pattern of fluctuations can be observed in other methods as well, where some excel on individual tasks but fail to maintain strong performance. In contrast, DOTS continues to deliver consistently high accuracy across various models and datasets. The stability of our method is attributed to its ability to dynamically select appropriate reasoning trajectories. The results indicate that DOTS is better suited to meet the demands of diverse tasks, demonstrating stronger robustness and generalization, making it a more reliable and adaptable approach for handling a wide variety of OOD challenges.

## 3.6 ABLATION STUDY

In this section, we perform the ablation study and assess the effectiveness of each component of our method: (1) **Without Searching:** To demonstrate the effectiveness of searching for the optimal action trajectory, we test the performance of the LLM tuned with a randomly selected action trajectory; (2) **Without Explanation:** To understand if training the planner to generate an explanation for

Table 5: Accuracy (%) on out-of-distribution (OOD) tasks.

| Method | DeepMind-Math | MMLU-pro | StrategyQA | DROP | Average |
|---|---|---|---|---|---|
| **External Planner: Finetuned Llama-3-8B-Instruct; Solver: Llama-3-70B-Instruct** | | | | | |
| CoT | 54.6 | 60.6 | 81.3 | 66.1 | 65.6 |
| LTM | 55.6 | **60.9** | **81.9** | 64.3 | 65.6 |
| PA | 58.1 | 54.2 | 80.3 | 58.7 | 62.8 |
| PoT | 73.0 | 57.3 | 74.8 | 62.8 | 66.9 |
| Self-refine | 73.9 | 59.5 | 77.8 | 64.8 | 69.0 |
| **DOTS: External** | **74.1** | 59.4 | 80.3 | **66.3** | **70.0** |
| **External Planner: Finetuned Llama-3-8B-Instruct; Solver: GPT4o-mini** | | | | | |
| CoT | 80.2 | **61.7** | 78.8 | 65.8 | 71.6 |
| LTM | 80.6 | 61.4 | **80.9** | 64.5 | 71.8 |
| PA | 82.2 | 48.1 | 78.3 | 67.0 | 68.9 |
| PoT | **87.7** | 57.1 | 77.9 | 72.4 | 73.7 |
| Self-refine | 85.9 | 58.3 | 77.2 | 72.3 | 73.4 |
| **DOTS: External** | 87.6 | 61.5 | 78.8 | **73.8** | **75.4** |
| **Solver: Finetuned Llama-3-8B-Instruct** | | | | | |
| CoT | 28.3 | 37.2 | **72.7** | 52.9 | 47.8 |
| LTM | 30.9 | 38.6 | 70.7 | **55.2** | 48.9 |
| PA | 29.3 | 34.5 | 69.7 | 51.6 | 46.3 |
| PoT | 48.1 | 37.3 | 63.9 | 44.6 | 48.5 |
| Self-refine | 44.9 | 33.1 | 65.3 | 47.1 | 47.6 |
| Vanilla SFT | 39.6 | **40.3** | 71.8 | 49.0 | 50.2 |
| **DOTS: Internalized** | **55.3** | 39.7 | 68.2 | 48.8 | **53.0** |

Table 6: Ablation Study

| | MATH | BBH | Game24 | TheoremQA | Average |
|---|---|---|---|---|---|
| **External Planner: Llama-3-8B-Instruct; Solver: GPT-4o-mini** | | | | | |
| DOTS: External | 75.4 | 84.2 | 65.2 | 41.4 | 66.5 |
| -w/o Searching | 69.2 | 78.6 | 28.9 | 40.2 | 54.2 |
| -w/o Explanation | 68.2 | 81.3 | 57.4 | 36.4 | 60.8 |
| **Internalized Planner & Solver: Llama-3-8B-Instruct** | | | | | |
| DOTS: Internalized | 34.4 | 69.7 | 21.9 | 16.1 | 35.5 |
| -w/o Searching | 31.4 | 55.8 | 19.6 | 15.1 | 30.5 |
| -w/o Explanation | 33.8 | 65.8 | 18.6 | 15.7 | 33.4 |

the optimal reasoning trajectory is helpful, we test DOTS's performance when the planner is trained to predict the trajectory without explanation.

The results in Table 6 indicate that both optimal trajectory searching and explanation generation are crucial in DOTS. For example, in the Game of 24, the planner trained without searching for the optimal trajectory did not consistently select the PoT action (which was considered the most effective for this task) in its trajectory. Additionally, we observe that without explanations, the planner's ability to predict optimal trajectories becomes less reliable. Incorporating explanations effectively guides the planner to learn to predict suitable action trajectories for the given questions.

### 3.7 OPTIMAL TRAJECTORY ANALYSIS FOR DIFFERENT TASKS

Table 7 shows the distribution of actions selected in the optimal trajectories by our planner on the MATH test set. The distribution suggests two key findings:

**DOTS adapts to the characteristics of specific questions**  In mathematics, number theory problems are more suitable to be solved with programs, so the proportion of PoT is higher, while geometry problems are not easily represented and solved with naive Python code; as a result, our planner mainly uses CoT for such problems. This indicates that DOTS tailors its action selection based on the unique characteristics of each problem type.

**DOTS adapts to the capability of specific task-solving LLMs**  As shown in Table 3, on the MATH dataset, GPT-4o-mini performs better using CoT for problem-solving, whereas Llama3-70B-instruct performs better using PoT. When GPT-4o-mini is the task-solving LLM, our fine-tuned planner selects a higher proportion of CoT actions; when Llama3-70B-Instruct is used, PoT actions dominate. This suggests that our planner is not only aware of the problem type but also adapts the reasoning action trajectory prediction based on the capabilities of the task-solving LLM.

Table 7: Planning action distributions of DOTS over three different layers on the MATH test set.

| Sub-tasks on MATH | Accuracy (%) | Analysis Layer | | | Solution | | Verification | |
|---|---|---|---|---|---|---|---|---|
| | | Rewr. | Deco. | Empty | CoT | PoT | Veri. | Empty |
| **External Planner: Llama-3-8B-Instruct; Solver: GPT-4o-mini** | | | | | | | | |
| Algebra | 92.1 | 0.03 | 0.05 | 0.92 | 0.90 | 0.10 | 0.29 | 0.71 |
| Prealgebra | 88.6 | 0.03 | 0.01 | 0.96 | 0.79 | 0.31 | 0.21 | 0.79 |
| Number Theory | 81.8 | 0.01 | 0.01 | 0.98 | 0.43 | 0.57 | 0.15 | 0.85 |
| Counting and Probability | 76.8 | 0.08 | 0.06 | 0.84 | 0.78 | 0.32 | 0.30 | 0.70 |
| Geometry | 61.8 | 0.03 | 0.01 | 0.96 | 0.95 | 0.05 | 0.06 | 0.94 |
| Intermediate Algebra | 57.1 | 0.05 | 0.02 | 0.93 | 0.85 | 0.15 | 0.44 | 0.56 |
| Precalculus | 52.6 | 0.06 | 0.02 | 0.92 | 0.95 | 0.05 | 0.46 | 0.54 |
| **External Planner: Llama-3-8B-Instruct; Solver: Llama-3-70B-Instruct** | | | | | | | | |
| Algebra | 74.9 | 0.03 | 0.04 | 0.93 | 0.77 | 0.23 | 0.12 | 0.88 |
| Prealgebra | 74.5 | 0.02 | 0.03 | 0.95 | 0.57 | 0.43 | 0.10 | 0.90 |
| Number Theory | 69.9 | 0.01 | 0.01 | 0.98 | 0.32 | 0.68 | 0.13 | 0.87 |
| Counting and Probability | 55.4 | 0.04 | 0.02 | 0.94 | 0.59 | 0.41 | 0.11 | 0.89 |
| Geometry | 39.6 | 0.05 | 0.01 | 0.94 | 0.76 | 0.24 | 0.18 | 0.82 |
| Precalculus | 36.9 | 0.07 | 0.03 | 0.90 | 0.78 | 0.22 | 0.28 | 0.72 |
| Intermediate Algebra | 34.6 | 0.03 | 0.01 | 0.96 | 0.72 | 0.28 | 0.20 | 0.80 |

Furthermore, we observe that question rewriting and decomposition were selected with a low frequency. This is likely because the MATH dataset consists of precise problems that do not benefit from rewriting. Additionally, given the strong reasoning abilities of Llama3-70B-Instruct and GPT-4o-mini, their CoT process inherently includes task decomposition, reducing the need for further planning interventions.

## 3.8 ADDITIONAL ANALYSES

**Few-shot In-context Learning Setting** Our main results report the performance with zero-shot evaluation. In cases where reasoning tasks are known in advance, a common approach to leveraging training data and improving the performance of closed-source LLMs is few-shot in-context learning (ICL), where training examples are incorporated directly into the context. Our external planner tuning can

Table 8: External planner tuning under the few-shot setting with GPT-4o-mini as the solver.

| Method | MATH | BBH | TheoremQA | Average |
|---|---|---|---|---|
| CoT | 72.3 | 84.2 | 38.2 | 64.9 |
| LTM | 72.7 | 83.4 | 37.3 | 64.5 |
| PA | 71.3 | 83.3 | 38.7 | 64.4 |
| PoT | 69.8 | 82.1 | 36.4 | 62.8 |
| Self-refine | 73.2 | 83.1 | 35.4 | 63.9 |
| **DOTS** | **75.4** | **86.1** | **39.9** | **67.1** |

also be utilized in this scenario seamlessly. Specifically, we can first construct few-shot ICL prompts for each potential reasoning action trajectory. Once the external planner selects the appropriate reasoning actions, the corresponding few-shot prompt will be chosen and applied. We evaluate the external planner tuning setup of DOTS, with Llama-3-8B-Instruct being the external planner and GPT-4o-mini being the solver LLM, in this setting. We compare our approach with the same baselines similarly implemented in the few-shot ICL setting, where we randomly selected 8 examples from MATH, 4 examples from each category of BBH, and 4 examples from TheoremQA to form the prompt.[3] All few-shot demonstrations were generated by GPT-4o and manually verified for quality.

As shown in the Table 8, DOTS continues to outperform baseline models. Interestingly, compared to Table 3, which presents the zero-shot results, adding few-shot demonstrations to static prompting methods does not lead to consistent improvement, except on the BBH dataset. This indicates that simply expanding the context with additional demonstrations does not always serve as an effective way to leverage available training data. In contrast, our method demonstrates its superior ability to effectively utilize the training data.

**How efficient is DOTS?** We compare the cost efficiency, measured by the average output token count, of each method (based on Llama-3-8B-Instruct) in Table 9. The result shows that DOTS consumes fewer tokens on average than other advanced approaches and only more than CoT. Advanced prompt engineering methods often introduce supplementary text to facilitate reasoning. However, not all ques-

Table 9: Avg. number of output tokens for each method (solver: Llama-3-8B-Instruct).

| Method | Avg. # of Output Tokens |
|---|---|
| CoT (Wei et al., 2022) | 263.6 |
| LTM (Zhou et al., 2023) | 436.4 |
| Self-refine (Madaan et al., 2024) | 527.6 |
| **DOTS: Internalized** | 409.1 |

---

[3]We excluded the "Game of 24" task because knowing the task in advance enables it to be solved with a straightforward program.

tions require this additional context to the same extent. By constructing training data via searching, our goal is to optimize the balance between minimizing extraneous steps and maintaining a high success rate, thereby reducing unnecessary output tokens. Our method avoids redundant reasoning actions, resulting in a more efficient system.

**Do we need more reasoning steps for difficult questions?** Recent research suggests that LLMs can better solve difficult questions by increasing the thinking time in the inference stage (Brown et al., 2024; OpenAI, 2024). In our study, we explore the relationship between question difficulty and the average reasoning action trajectory length. The trajectory length is determined by assigning a value of $0$ to the EMPTY module and $1$ to all other actions, while the question difficulty is derived from annotated levels on the MATH dataset. Figure 3 presents that harder problems demand more computational steps, resulting in longer reasoning trajectories. Case analyses further reveal that our planner increases the proportion of verification steps as problem difficulty

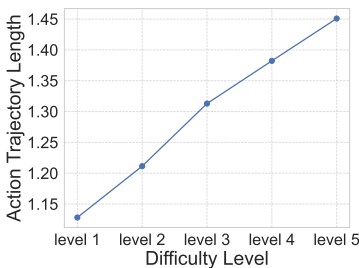

Figure 3: Average reasoning trajectory length per difficulty level on MATH for DOTS (solver: GPT-4o-mini; External planner: Llama3-8B-Instruct).

rises. This highlights an exciting fact — LLMs can learn to employ more reasoning steps for challenging problems through exploration, without requiring explicit expert guidance.

## 4    RELATED WORK

**Prompt engineering for LLM reasoning** LLMs have demonstrated remarkable proficiency in solving complex reasoning tasks (Lewkowycz et al., 2022). The Chain-of-Thought (CoT) approach, introduced by Wei et al. (2022), significantly improves performance on reasoning problems by prompting LLMs to think step-by-step in natural language. Chen et al. (2022) and Gao et al. (2023) proposed the Program-of-Thought prompting method, where code is used as an intermediate reasoning step. Advanced prompt engineering methods, such as question decomposition (Zhou et al., 2023) and self-verification (Madaan et al., 2024), have also proven effective in improving reasoning performance. Additionally, recent approaches have incorporated automatic prompt optimization based on training data. For instance, Wang et al. (2023) refines prompts by analyzing error cases, and self-discovery (Zhou et al., 2024) utilizes modular reasoning components to construct the task-adaptive prompt. However, these automated prompt optimization techniques still produce static prompts for all instances. Srivastava et al. (2024) proposed instance-level prompt optimization via LLM self-refining, but it relies on expert-designed workflows and lacks active adaptation. In our method, we internalize the reasoning action selection capability into the LLM itself without an expert-designed workflow, allowing it to autonomously fit both the characteristics of questions and the inherent capability of task-solving LLM.

**Searching for boosting LLM reasoning** Recent research suggests that incorporating searching mechanisms can significantly enhance LLM reasoning. In the inference process, Tree-of-Thought (ToT) (Yao et al., 2024) and Graph-of-Thought (GoT) (Besta et al., 2024) have been proposed to search and investigate different reasoning paths, either by leveraging the LLM itself (Yao et al., 2024) or designing heuristic functions (Hao et al., 2023) as the signal to evaluate each step. More recently, Monte Carlo Tree Search (MCTS) has been introduced to assist the LLM in learning how to evaluate each step (Qi et al., 2024; Xie et al., 2024). The searching mechanism can also be used in training to collect training instances for improving LLM reasoning (Luo et al., 2024). However, all these searching methods treat each "CoT reasoning step" as the atomic component or step in searching, while we choose each reasoning action as the atomic component in our case.

## 5    CONCLUSION

In this paper, we introduce DOTS, a method that enables LLMs to autonomously think about appropriate reasoning actions before answering questions. By defining atomic reasoning action modules, searching for optimal action trajectories, and training LLMs to plan for reasoning questions, we enable LLMs to dynamically adapt to specific questions and their inherent capability. The flexibility of our two learning paradigms, i.e., external and internalized planner tuning, further highlights the adaptability of our method to different LLMs. Our experimental results show the effectiveness of DOTS, revealing the promise of harnessing explorations and evaluations to turn LLMs into planners for better reasoning.

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

## A    TRAINING IMPLEMENTATION

In our optimal trajectory search, we set the number of iterations ($K$) to 2 and the number of evaluation times ($N_{eval}$) to 4. The number of paths retained $N_1$ is set to 8 and $N_2$ to 3. Throughout the search, we maintain a sampling temperature of 0.4. Searching on the training datasets eventually yields 1722 for GPT-4o-mini, 1624 for Llama-3-70B-Instruct, and 2140 for Llama-3-8B-Instruct training examples for planner tuning, respectively. We up-sample few-shot examples by a factor of 8 and fine-tune the planner LLM using the LitGPT library (AI, 2023), applying a learning rate of 2e-5, a global batch size of 64, a maximum sequence length of 4096, and training epoch of 4.

## B    CASE STUDY

In this section, $Q_I$ is a numerical reasoning problem that can be solved with Python code easily, so our tuned LLM selected PoT to solve it. $Q_{II}$ is a question widely tested in the community where GPT-4o even tends to make mistakes, but after tuning with searched trajectory data, the LLM actively chose to use code to solve the problem.

---

**Question 1 Case Study**

**Question I:**
A book with 50 pages numbered 1 through 50 has its pages renumbered in reverse, from 50 to 1. For how many pages do both sets of page numbers share the same digit?

**Answer from GPT-4o-2024-08-06 (Incorrect):**
To solve this problem, we need to determine how many pages have the same ones digit in both the original numbering (1 to 50) and the reversed numbering (50 to 1).
First, let's understand the problem: - The original numbering of the pages is from 1 to 50. - The reversed numbering is from 50 to 1, meaning page 1 becomes 50, page 2 becomes 49, and so on, until page 50 becomes 1. We need to find pages where the ones digit of the original page number is the same as the ones digit of the reversed page number. Let's denote the original page number as $n$ and the reversed page number as $51 - n$. We are looking for $n$ such that the ones digit of $n$ is equal to the ones digit of $51 - n$.
The ones digit of a number is the remainder when the number is divided by 10. Therefore, we need:

$$n \equiv 51 - n \pmod{10}$$

Since $51 \equiv 1 \pmod{10}$, we have:

$$2n \equiv 1 \pmod{10}$$

Thus, multiplying both sides of the congruence by 5 gives:

$$n \equiv 5 \times 1 \equiv 5 \pmod{10}$$

This means $n$ must have a ones digit of 5. The pages with a ones digit of 5 in the original numbering are 5, 15, 25, 35, and 45.
Thus, the number of pages that share the same one's digit in both numbering systems is: 5

**Answer from our internal planner tuned Llama3-8b-Instruct:**
It can efficiently iterate through the range of page numbers to count occurrences of the same ones digit. Programming solver provides a quick and reliable method to handle the counting of digits across a defined range.
Required Action: programming

```
count = 0
for original_page in range(1, 51):
    renumbered_page = 51-original_page
    # Check if the ones digit is the same
    if original_page % 10 == renumbered_page % 10:
        count += 1
print(count)
```

After execution, we get: 0
Answer: 0

---

| Sub Task | CoT | PoT | Dots |
|---|---|---|---|
| object counting | 94.4 | 98.0 | **98.0** |
| temporal sequences | 96.8 | 83.6 | **97.2** |
| tracking shuffled objects five objects | 95.2 | 78.8 | **96.4** |
| word sorting | 50.8 | 98.0 | **98.0** |
| salient translation error detection | 61.6 | 59.2 | **62.6** |
| penguins in a table | 71.2 | 54.8 | **73.3** |
| snarks | 83.1 | 69.1 | **83.3** |
| ruin names | 81.2 | 80.2 | **81.2** |
| web of lies | 91.2 | 59.2 | **92.0** |
| navigate | 96.4 | 90.4 | **96.8** |
| date understanding | **80.8** | 52.8 | 76.4 |
| hyperbaton | **92.0** | 67.6 | 91.0 |
| dyck languages | 38.0 | 42.4 | **54.4** |
| tracking shuffled objects three objects | 99.2 | 89.2 | **99.2** |
| formal fallacies | 82.0 | 74.4 | **82.0** |
| tracking shuffled objects seven objects | 91.2 | 73.2 | **92.8** |
| causal judgement | **62.6** | 62.6 | 62.0 |
| sports understanding | 85.6 | 74.8 | **87.6** |
| logical deduction five objects | 85.6 | 80.4 | **88.8** |
| movie recommendation | 62.0 | 56.0 | **62.8** |
| logical deduction three objects | 99.2 | 96.4 | **99.2** |
| multistep arithmetic two | 98.0 | 100.0 | **100.0** |
| boolean expressions | 99.2 | 96.8 | **99.2** |
| geometric shapes | 56.4 | 65.6 | **78.8** |
| disambiguation qa | 42.8 | 40.8 | **45.6** |
| logical deduction seven objects | 76.8 | 71.6 | **82.0** |
| reasoning about colored objects | 87.6 | 66.0 | **87.6** |
| **Average** | 80.3 | 73.9 | **84.2** |

Table 10: Results of BBH sub-tasks.

---

**Question 2 Case Study**

**Question II:**
How many r in word 'strawberry'?

**Answer from GPT-4o-2024-08-06 (Incorrect):**
The word "strawberry" contains 2 letter "r"s.

**Answer from our internal planner tuned Llama-8b-Instruct:**
It can efficiently count the occurrences of a specific character (in this case, 'r') in a word. Programming_solver provides a quick and reliable method for counting characters in text, ensuring accuracy in the result.
Required Action: programming

```
word = 'strawberry'
count=word.lower().count('r')
print(count)
```

After execution, we get: 3
Answer: 3

---

## C  SUB-TASK PERFORMANCE OF BBH

Table 10 shows the results of BBH sub-tasks of GPT-4o-mini as task-solving LLM and our tuned Llama-3-8B-instruct as the planner. From the table, we can see that our planner has learned to select appropriate actions based on the task. For instance, in the word sorting task, our model consistently

uses code to solve the problem. Additionally, for the Dyck languages task, our method outperforms both CoT and PoT. This is because the task is easy to verify, and our model proactively requests GPT-4o-mini to verify the answer, thereby improving performance. The only exception is the date understanding task. Upon analysis, we found that while code could solve this type of problem using Python's "datetime" library, it often fails to follow the required output format. This leads to lower accuracy in our method's prediction when choosing writing a program as the reasoning action.

## D  PROMPTS USED IN EXPERIMENTS

**Prompt for query rewrite module**

In this step, you need to reveal the Core Question with only a simple sentence and useful information. The output follows the format:
core question:...
Note: Please extract the question-solving information related to the problem, and list them one by one.
useful information:...

**Prompt for query decomposition module**

In this step, you need to reflect on the problem, and describe it in your own words. Analyze how you can decompose the problem into smaller, more manageable sub-tasks. Pay attention to small details, nuances, notes and examples in the problem description.

**Prompt for CoT module**

In this step, you need to think step by step with words, solve the problem and get the answer.

**Prompt for PoT module**

In this step, you need to write Python codes to solve the query. Use the simplest and most straightforward programming methods to solve the problem. For instance, if a query can be efficiently solved using a brute force method, prefer it over heuristic or more complex methods. Utilize any available and commonly-used libraries that can simplify the task or improve code maintainability. All the calculations must leverage codes. Print out the results with the print() function. Before executing the program, you have no idea of the final answer. Don't show it in your comment or code. And don't use the plot function.
In this step, start with "# Now write Python codes to answer this question and use print() to print out the result"

**Prompt for self-verification module**

In this step, you need to carefully verify the correctness of the previous thoughts with natural language. You need to formulate a verification question (not the same question as before) based on the final answer and then verify the final answer you have. If the results are incorrect, the last line should end up with "The answer is: incorrect". Otherwise, the last line should end with "The answer is: correct"

---

**Prompt for explanation generation**

**Action Categories**:
1. Understanding process: query rewriting: Rewrite the question and answer it. Decomposition: Decompose the questions into multiple subtasks to solve the sub-question. 2. Solving process: chain of thought: For step-by-step reasoning with language. programming: For programming solver. 3. Verification process: self-verification: To check the correctness of the solution.

**Task Instruction**: For the given question, explain why the above Required actions are necessary.

**Example 1:**

Query: Find $2 \cdot 5^{-1} + 8 \cdot 11^{-1} \pmod{56}$. Express your answer as an integer from $0$ to $55$, inclusive.

Required Action: programming, self-verification

Explanation: This is a Modular arithmetic problem. The problem can be solved using straightforward python code with sympy library, particularly modular arithmetic. Besides, this type of problem is relatively easy to verify. After computing the result, one can check the calculations step by step to ensure correctness and verify that the final answer is within the given range (0 to 55 inclusive). Programming solver is more efficient and accurate for this type of calculation and the verifier ensures the correctness of the result and adherence to the given constraints.

... **(multiple examples)**

Query: Given Query

Required Action: Actions After Searching

Explanation:

---

---

**Prompt for prior knowledge exploration module**

...(test query)

Before solving the problem, take a moment to: Choose the most appropriate action trajectories from: 1. Understanding Process:

- query rewriting: - Suitable for: Ambiguous or poorly phrased questions. - Reasoning: This action is ideal when the question is not clear or is open to multiple interpretations. By rewriting the question, we can clarify the intent and ensure that the answer addresses the correct issue. This is particularly useful in natural language processing tasks where the original query might be vague or contain errors.

-Decomposition: - Suitable for: Complex questions that can be broken down into simpler, more manageable parts. - Reasoning: When a question involves multiple components or steps, decomposing it into subtasks can make it easier to tackle. This approach is beneficial for multi-step problems, such as those found in project management, research planning, or complex problem-solving scenarios where each subtask can be addressed individually before integrating the solutions.

2. Solving Process: - chain of thought: - Suitable for: Questions requiring logical reasoning or step-by-step deduction. - Reasoning: This method is effective for problems that benefit from a structured approach, such as mathematical proofs, logical puzzles, or any scenario where a sequential thought process is necessary to arrive at the correct answer. It allows for a clear demonstration of the reasoning path taken to solve the problem.

- programming: - Suitable for: Questions that can be solved algorithmically or require computational solutions. - Reasoning: When a problem can be addressed through coding, such as data analysis, algorithm design, or simulation tasks, using a programming solver is appropriate. This approach is particularly useful for problems that can be solved with brute force searching, where writing and executing code can efficiently produce the desired outcome.

3. Verification Process: - verifier: - Suitable for: Questions where the correctness of the solution is critical. - Reasoning: This action is essential when the accuracy of the solution must be confirmed, such as in scientific research, engineering calculations, or any high-stakes decision-making process. Verification ensures that the solution is not only plausible but also correct, providing confidence in the results.

You need to first think about the question and select the best action trajectories, for example, [programming, verifier] to solve it. Then proceed with solving the problem using your selected strategies. Remember to follow the question format.

## E  MORE BASELINES

### E.1  THREE-LAYERS STATIC PROMPTING

The baselines in experiments did not utilize all layers. To ensure a fairer comparison, in this section, we compared our method with results using all layers, as shown in Table 11. The results demonstrate that our method still outperforms all baselines.

### E.2  HYBRID TRAINING

The second baseline is about hybrid training, i.e., using a mixture of training data with both CoT and PoT (Yue et al., 2023). We use the same training instance and each instance is provided with both the PoT and CoT reasoning paths. We reported the results in Table 12. The result shows that our method still outperforms the hybrid training. From the results, it can also be observed that for datasets where CoT outperforms PoT, training solely with CoT yields better performance. We analyzed several cases and found that although mixed training allows the LLM to choose between CoT and PoT dynamically, the results tend to perform random selection rather than actively analyzing the problem and selecting the most suitable path for the given question carefully because we didn't guide the LLM learn this. This highlights the importance of searching for the optimal reasoning path before tuning the LLM.

| Method | MATH | BBH | Game 24 | Theorem QA | DM math | MMLU pro | Strategy QA | DROP | Average |
|---|---|---|---|---|---|---|---|---|---|
| Task solving LLM: Llama3-70b-Instruct | | | | | | | | | |
| Rewr.+CoT+Veri. | 50.9 | 73.1 | 28.2 | 28.8 | 58.1 | 60.1 | **81.9** | 66.0 | 55.9 |
| Rewr.+PoT+Veri. | 55.3 | 70.5 | 67.9 | 30.6 | 73.4 | 58.1 | 79.1 | 65.2 | 62.5 |
| Deco.+CoT+Veri. | 51.1 | 74.2 | 29.6 | 29.1 | 58.6 | **60.5** | 81.4 | 65.7 | 56.3 |
| Deco.+PoT+Veri. | 55.8 | 72.5 | **68.1** | 30.8 | 73.9 | 58.6 | 78.9 | 65.6 | 63.0 |
| DOTS | **57.7** | **77.3** | 67.7 | **31.2** | **74.1** | 59.4 | 80.3 | **66.3** | **64.3** |
| Task solving LLM: GPT-4o-mini | | | | | | | | | |
| Rewr.+CoT+Veri. | 71.9 | 79.5 | 32.1 | 36.9 | 80.1 | 61.1 | 80.6 | 68.5 | 63.8 |
| Rewr.+PoT+Veri. | 73.4 | 76.7 | 67.9 | 34.8 | 85.8 | 58.4 | 75.8 | 72.9 | 68.2 |
| Deco.+CoT+Veri. | 72.7 | 79.8 | 33.5 | 37.8 | 82.8 | **62.3** | **81.2** | 67.3 | 64.7 |
| Deco.+PoT+Veri. | 73.6 | 78.7 | **68.7** | 35.2 | 86.5 | 59.3 | 76.9 | 71.9 | 68.9 |
| DOTS | **75.4** | **84.2** | 65.2 | **41.4** | **87.6** | 61.5 | 78.8 | **73.8** | **71.0** |
| Task solving LLM: Finetuned Llama3-8b-Instruct | | | | | | | | | |
| Rewr.+CoT+Veri. | 29.5 | 48.9 | 14.5 | 15.8 | 30.1 | 38.7 | 70.3 | **52.9** | 37.6 |
| Rewr.+PoT+Veri. | 27.2 | 48.3 | 17.8 | 14.9 | 46.9 | 35.5 | 64.9 | 47.1 | 37.8 |
| Deco.+CoT+Veri. | 30.1 | 49.5 | 15.5 | 14.6 | 31.7 | 39.2 | **71.3** | 52.1 | 38.0 |
| Deco.+PoT+Veri. | 27.6 | 49.2 | 18.9 | 15.2 | 47.8 | 36.2 | 64.7 | 45.8 | 38.2 |
| DOTS | **34.4** | **69.7** | **21.9** | **16.1** | **55.3** | **39.7** | 68.2 | 48.8 | **44.3** |

Table 11: The three-layer baselines performance

| SFT Method | MATH | BBH | Game24 | Theorem QA | DM math | MMLU pro | Strategy QA | DROP | Average |
|---|---|---|---|---|---|---|---|---|---|
| Vanilla SFT | 33.9 | 61.0 | 18.5 | 14.8 | 39.6 | **40.3** | **71.8** | **49.0** | 36.7 |
| Hybrid SFT | 32.3 | 53.3 | 18.7 | 15.6 | 49.1 | 36.5 | 65.9 | 46.2 | 39.7 |
| DOTS | **34.4** | **69.7** | **21.9** | **16.1** | **55.3** | 39.7 | 68.2 | 48.8 | **44.3** |

Table 12: The result of hybrid training

# F DISTRIBUTION OF REASONING PATH

In this section, we show a detailed distribution of reasoning paths. We denote the reasoning paths as:

- P1: Empty, CoT, Empty
- P2: Empty, PoT, Empty
- P3: Empty, CoT, verification
- P4: Empty, PoT, verification
- P5: Question Decomposition, CoT, Empty
- P6: Question Decomposition, PoT, Empty
- P7: Question Decomposition, CoT, verification
- P8: Question Decomposition, PoT, verification
- P9: Question Rewrite, CoT, Empty
- P10: Question Rewrite, PoT, Empty
- P11: Question Rewrite, CoT, verification
- P12: Question Rewrite, PoT, verification

The distribution is shown in Table 13. We can still maintain the conclusion drawn from the section 3.7. For different tasks, the planner predicts different action trajectories. For example, the planner selects PoT more frequently in Game of 24, while preferring CoT in DROP. The planner can also adapt to the LLM ability. For instance, GPT-4o-mini excels at solving problems using CoT, leading to a higher selection rate of CoT compared to Llama-3-70B-Instruct.

| Dataset | P1 | P2 | P3 | P4 | P5 | P6 | P7 | P8 | P9 | P10 | P11 | P12 |
|---------|-----|-----|-----|-----|-----|-----|-----|-----|-----|-----|-----|-----|
| Task solving LLM: Llama3-70B-Instruct | | | | | | | | | | | | |
| MATH | **50.5** | **26.0** | **8.4** | 3.7 | 3.1 | 1.1 | 0.4 | 1.9 | 1.7 | 1.9 | 1.3 | 0.0 |
| BBH | 32.1 | 16.2 | **21.1** | 9.1 | **10.9** | 0.3 | 0.5 | 0.7 | **9.0** | 0.1 | 0.0 | 0.0 |
| Game 24 | 0.2 | 0.7 | **4.3** | 94.1 | 0.0 | 0.4 | 0.0 | 0.3 | 0.0 | 0.0 | 0.0 | 0.0 |
| TheoremQA | **42.6** | **30.3** | **13.0** | 8.1 | 0.4 | 0.0 | 5.0 | 0.0 | 0.4 | 0.2 | 0.0 | 0.0 |
| Deepmind Math | 1.1 | 32.1 | 6.1 | **59.1** | 0.2 | 0.2 | 0.0 | 0.6 | 0.0 | 0.0 | 0.0 | 0.6 |
| MMLU-pro | **59.8** | 5.2 | **31.0** | 0.4 | 1.0 | 0.4 | 2.0 | 0.2 | 0.0 | 0.0 | 0.0 | 0.0 |
| StrategyQA | **62.6** | 4.0 | **20.1** | 2.1 | **10.2** | 0.0 | 1.0 | 0.0 | 0.0 | 0.0 | 0.0 | 0.0 |
| DROP | **82.8** | 0.9 | **16.2** | 0.0 | 0.1 | 0.0 | 0.0 | 0.0 | 0.0 | 0.0 | 0.0 | 0.0 |
| Task solving LLM: GPT4o-mini | | | | | | | | | | | | |
| MATH | **54.9** | **12.4** | **20.1** | 5.1 | 1.6 | 1.1 | 0.6 | 0.0 | 3.4 | 0.0 | 0.8 | 0.0 |
| BBH | **45.4** | **15.8** | **14.7** | 2.0 | **13.0** | 3.7 | 0.0 | 0.0 | **5.0** | 0.0 | 0.0 | 0.0 |
| Game 24 | 0.5 | 1.6 | 0.3 | 97.6 | 0.0 | 0.0 | 0.0 | 0.0 | 0.0 | 0.0 | 0.0 | 0.0 |
| TheoremQA | **43.8** | **25.2** | **28.4** | 0.0 | 1.1 | 0.0 | 0.0 | 0.0 | 1.5 | 0.0 | 0.0 | 0.0 |
| Deepmind Math | 2.1 | **41.1** | 0.6 | **50.2** | 0.0 | 0.0 | 0.0 | 0.0 | 0.0 | 4.0 | 0.0 | 2.0 |
| MMLU-pro | **71.3** | **20.7** | 5.0 | 0.0 | 0.0 | 2.0 | 0.0 | 0.0 | 0.8 | 0.0 | 0.2 | 0.0 |
| StrategyQA | **68.1** | 7.0 | **19.1** | 0.0 | 5.7 | 0.1 | 0.0 | 0.0 | 0.0 | 0.0 | 0.0 | 0.0 |
| DROP | **98.7** | 1.0 | 0.2 | 0.0 | 0.0 | 0.1 | 0.0 | 0.0 | 0.0 | 0.0 | 0.0 | 0.0 |

Table 13: The action trajectory distribution of all datasets

| Method | MATH | BBH | Game 24 | Theorem QA | DM math | MMLU pro | Strategy QA | DROP | Average |
|--------|------|-----|---------|-----------|---------|----------|-------------|------|---------|
| Solver: GPT4o-mini | | | | | | | | | |
| DOTS | **75.4** | **84.2** | **65.2** | **41.4** | **74.1** | **59.4** | 80.3 | **66.3** | **68.4** |
| w/o Searching | 69.2 | 78.6 | 28.9 | 40.2 | 62.5 | 58.3 | 79.5 | 64.5 | 60.2 |
| w/o Explanation | 68.2 | 81.3 | 57.4 | 36.4 | 58.5 | 59.2 | **81.2** | 66.1 | 63.5 |
| Solver: Llama3-8B-Instruct | | | | | | | | | |
| DOTS | **34.4** | **69.7** | **21.9** | **16.1** | **55.3** | **39.7** | 68.2 | 48.8 | **44.3** |
| w/o Searching | 31.4 | 55.8 | 19.6 | 15.1 | 42.5 | 38.2 | 67.5 | 47.5 | 39.7 |
| w/o Explanation | 33.8 | 65.8 | 18.6 | 15.7 | 47.2 | 38.5 | **69.6** | **50.2** | 42.4 |

Table 14: Full result of ablation study

# G  ABLATION STUDY FULL RESULT

In this section, we provide the full ablation result of all datasets in Table 14. It reveals the effectiveness of each step of our method.

# H  CAN WE ADD PRIOR KNOWLEDGE IN THE PROMPT TO HELP THE LLM SELECT ACTIONS?

We designed an experiment with Llama-3-8B-Instruct, which shows that simply feeding the prior knowledge as instruction is insufficient for effective planning. Specifically, we compared the internalized DOTS planner with two variants: (1) Vanilla Prompting-based Planning, where an off-the-shelf Llama3-8B-Instruct model was directly prompted to select optimal actions for a given question based on its parametric knowledge; and (2) Prompting-based Planning w/ Prior Knowledge, where the model was prompted with the prior knowledge that we summarized from the solver LLM's optimal trajectories (e.g., common rules for action selection) as part of the instructions at test time. The prompt is provided in D. In both variants, we did not train the planner LLM. The results are shown in Table 15. Our experimental results show that feeding no prior knowledge, or simply feeding the prior knowledge without SFT, is insufficient for an LLM to plan effectively for reasoning tasks. We analyzed several cases and found that LLMs often failed in selecting the best trajectory. For example, one prior knowledge in the prompt is that "decomposition is useful when a question involves multiple components...". We find that prompt-based LLM tends to select a fixed subset of actions including decomposition. However, our search results indicate that incorporating decomposition does not provide significant benefits in tasks such as Game of 24. Because decomposition appears reasonable to the LLM, it is only through search and learning that the LLM can know that

| Method | MATH | BBH | Game of 24 | Theorem QA | Deepmind -math | MMLU -pro | SQA | DROP | Average |
|---|---|---|---|---|---|---|---|---|---|
| Vanilla Prompting -based Planning | 29.5 | 48.7 | 11.9 | 13.3 | 34.9 | 38.0 | 69.2 | 53.0 | 37.3 |
| Prompting-based Planning with Prior Knowledge | 30.2 | 47.5 | 13.0 | 14.5 | 34.2 | 37.8 | **69.3** | **55.2** | 37.7 |
| DOTS | **34.4** | **69.7** | **21.9** | **16.1** | **55.3** | **39.7** | 68.2 | 48.8 | **44.3** |

Table 15: The result of using prompt with prior knowledge

decomposition is unnecessary for this type of problem. It clearly demonstrates that relying solely on naive prompt design is insufficient for enabling LLMs to learn how to reason about selecting paths effectively. The reason is that enumerating all detailed and comprehensive prior knowledge in the prompt is challenging, as it is difficult to account for numerous case-by-case scenarios. Our method integrates prior knowledge from explanations with the searched optimal trajectory, effectively teaching the model to learn how to select the optimal trajectory. In our view, searching is to implicitly acquire the "prior knowledge" and training is a reliable way to make LLMs learn the "prior knowledge" to determine how to select reasoning actions.

