# OpenReview forum: "DOTS: Learning to Reason Dynamically in LLMs via Optimal Reasoning Trajectories Search"
_ICLR.cc/2025/Conference — ICLR 2025 Poster_

### Official Review · Reviewer_Npg3 · 2024-10-16

**Soundness:** 4
**Presentation:** 4
**Contribution:** 2
**Rating:** 6
**Confidence:** 4

**Summary:**

The author creates a planning prompt strategy named DOTS. It can choose the path to solve the problem via a small model (as the planner) and prompt the large model to solve the problem. The path contains 3 parts: the Analysis Layer, the Solution Layer, and the Solution Layer. The experiment shows that both external and internalized planner can help the large model to solve the problem more effectively and efficiently. The ablation study also shows that the searching and explanation in planner's plan is useful.

**Strengths:**

1. The paper is easy to read and follow. Its ideas are presented straightforwardly along with illustrations.

2. It simplifies the planning prompt construction process.

3. The authors study the DOTS over a wide range of benchmarks.

**Weaknesses:**

1. The comparison between the DOTS and other methods is not fair. The DOTS's prompt contains 3 layers, however, the baselines only contain 1/2 layer. The authors should compare the baseline with 3 layers (for example compulsory use of Decomposition + POT + self-verifications at the same time in MATH) to make the comparison more fair.

2. The method is not generalizable enough. It needs to SFT a new model for each large model / new atomic action in layers / new layers. (Although the authors claim that this is an advantage to make the prompt more suitable for a given model, it is still a limitation for the method.)

3. The improvement of the DOTS is not significant, especially in the OOD dataset.

4. The ablation study only shows the result in the IND benchmark

5. The ablation study in Table 6 does not explain the inherent reason that DOTS can improve performance, it's not enough for ICLR. (For example, can a large model have a similar performance with DOTS by only adding prior knowledge in the explanation?). (see question 3,4,6 below for more details)

**Questions:**

1. could you present the distribution of the path the planner will choose in the experiment? (i.e. x% of the path will choose Empty + POT + self-verifications, y % of the path will choose Decomposition + COT + Empty, etc.) (This is different from Table 7 in 2 parts: 1. It shows the distribution of the 12 types of combination of 3 layers in the path, 2. It needs to contain more benchmarks besides the MATH dataset)

2. Can 2 findings in Section 3.7 be generalized to other datasets?

3. (From weakness 1) The authors should add a comparison with the baseline with 3 layers (and the choice of the layer should be reasonable towards a given benchmark) to make the comparison more fair.

4. From my perspective, the OOD benchmark is more important than the IND benchmark, however, the authors only show the ablation study in the IND benchmark. Why?

5. The improvement of the DOTS seems to come from the explanation (if you compare the result in Table 6's "-w/o Explanation" line with the second to the last line in Tables 3 and 4). So, does the prior knowledge in the prompt cause the improvement?

6. (minor) The author claims that "the fine-tuned LLMs are constrained to follow the same reasoning format of the training data (e.g., CoT (Luo et al., 2023)) and lack the flexibility to adopt other reasoning strategies" (in line 53,77-78). However, FireAct[1], shows that the fine-tuned LLMs can change the reasoning strategies (COT or ReAct) by itself without any additional prompt. Maybe training COT and POT at the same time can help the model to be more flexible. The authors should revise this point to make it more accurate.

7. In Internalized setting, I'm curious that if llama3-8b is trained only in the Reasoning Process and Answer (like FireAct), what performance will be, will the trained model choose the Reasoning Process automatically? (It should also be a baseline), although I admit this can't be used in external setting.

8. (minor) In Table 2, "Few-shot" is not a kind of Distribution, maybe 4 columns (add example number column) can be more clear.

[1] FIREACT: TOWARD LANGUAGE AGENT FINE-TUNING

---

> ### Author Response · Authors · 2024-11-25
> **Reply to reviewer Npg3**
>
> Thank you very much for the feedback. We provide clarifications to your questions below.
>
> 1. W1 Q3: Please refer to our general reply about more baselines.
>
> 2. W2: We appreciate the reviewer's thoughtful feedback. (1) Generalization w.r.t. new LLM: We view the ability to adapt to the capabilities of different LLMs as a strength rather than a limitation for two key reasons. In practical applications, when developing a system, we typically begin by selecting a base LLM and then optimizing it. As a result, creating a planner that works universally across all LLMs is less meaningful than enhancing one particular LLM. Besides, for some questions, there may be no universal optimal trajectory. As demonstrated in Table 3, different LLMs exhibit varying performances even when using the same prompt engineering method. Thus, we argue that developing a universal planner may not be feasible. (2) Generalization w.r.t. new layers or actions: Our current pipeline cannot directly work with new layers or actions. When new layers or actions are added to the reasoning space, one will need to search for the optimal trajectories again, as now the search space has changed and the optimal trajectories in the past may not be optimal anymore. An LLM planner will then be trained based on the new set of optimal trajectories for the training instances. We argue that the additional work for this generalization is necessary (as we aim for optimality in planning), but the cost can be reduced. For example, the optimal trajectories from the past can be reused as results when the new layers or actions were not invoked when one decides whether the new layers or actions are helpful. Future work can also explore continual learning for efficiently adapting an existing LLM planner to plan with the new components.
>
> 3. W3: Please refer to our general reply about OOD task performance.
>
> 4. W4 Q4: We thank the reviewer for this suggestion and have added the full ablation result of all tasks in Appendix G. It consistently shows the importance of each module in our method.

---

> > ### Author Response · Authors · 2024-11-25
> > **Reply to reviewer Npg3**
> >
> > 5. W5 Q5: Thank you for your careful reading and insightful questions. Here are our thoughts about why our method is effective. Our approach has two main parts: searching for the optimal trajectory and generating explanations. For searching, LLMs can acquire knowledge about what action trajectory to choose for certain types of questions by learning from the optimal action trajectory acquired from searching. The importance of providing optimal trajectories is demonstrated in our ablation study and hybrid training in general reply sections. Without providing search-obtained optimal paths, LLMs would not effectively connect this problem pattern with its optimal action trajectory. For explanation generation, explanations aid the LLM's in learning. Our ablation study proves that including explanations leads to improvements. We believe this is because adding explanations reduces the reasoning burden of planning, and the associative words mentioned during the explanation process better help LLMs establish connections between question types and optimal trajectories. For example, through the analysis of the question content, the LLM recognizes it as an equation problem, and through this keyword "equation," it can more easily remind the LLM that the optimal path for solving the equation involves PoT and verification.
> >
> > For the question, “can a large model have a similar performance with DOTS by only adding prior knowledge in the explanation?” we designed an experiment with Llama-3-8B-Instruct, which shows that simply feeding the prior knowledge as instruction is insufficient for effective planning. Specifically, we compared the internalized DOTS planner with two variants: (1) Vanilla Prompting-based Planning, where an off-the-shelf Llama3-8B-Instruct model was directly prompted to select optimal actions for a given question based on its parametric knowledge; and (2) Prompting-based Planning w/ Prior Knowledge, where the model was prompted with the prior knowledge that we summarized from the solver LLM’s optimal trajectories (e.g., common rules for action selection) as part of the instructions at test time. In both variants, we did not train the planner LLM. The prompts are presented in Appendix H of the pdf.
> >
> > | Method                                   | MATH | BBH  | Game of 24 | TheoremQA | Deepmind-math | MML U-pro | Strategy QA | DROP | Average |
> > |-----------------------------------------|------|------|------------|-----------|---------------|-----------|-------------|------|---------|
> > | Vanilla Prompting-based Planning        | 29.5 | 48.7 | 11.9       | 13.3      | 34.9          | 38.0      | 69.2        | 53.0 | 37.3    |
> > | Prompting-based Planning with Prior Knowledge | 30.2 | 47.5 | 13.0       | 14.5      | 34.2          | 37.8      | 69.3        | 55.2 | 37.7    |
> > | DOTS                                    | 34.4 | 69.7 | 21.9       | 16.1      | 55.3          | 39.7      | 68.2        | 48.8 | 44.3    |
> >
> > Our experimental results show that feeding no prior knowledge, or simply feeding the prior knowledge without SFT, is insufficient for an LLM to plan effectively for reasoning tasks. We analyzed several cases and found that LLMs often failed in selecting the best trajectory. For example, one prior knowledge in the prompt is that “decomposition is useful when a question involves multiple components…”. We find that prompt-based LLM tends to select a fixed subset of actions including decomposition. However, our search results indicate that incorporating decomposition does not provide significant benefits in tasks such as Game of 24. Because decomposition appears as a reasonable action to solve this task to the LLM, it is only through search and learning that the LLM can know that decomposition is unnecessary for this type of problem. It clearly demonstrates that relying solely on naive prompt design is insufficient for enabling LLMs to learn how to reason about selecting paths effectively. The reason is that enumerating all detailed and comprehensive prior knowledge in the prompt is challenging, as it is difficult to account for numerous case-by-case scenarios. Our method integrates prior knowledge from explanations with the searched optimal trajectory, effectively teaching the model to learn how to select the optimal trajectory. In our view, searching is to implicitly acquire the “prior knowledge” and training is a reliable way to make LLMs learn the “prior knowledge” to determine how to select reasoning actions.

---

> > > ### Author Response · Authors · 2024-11-25
> > > **Reply to reviewer Npg3**
> > >
> > > 6. Q1, Q2, Q6, Q8: Thank the reviewer for this suggestion in paper writing. We have revised the pdf and (1) added the result of the path distribution for all the datasets in Appendix F. We could also maintain the same observation of the experiment sec 3.7. (2) revise the statement of related works in the introduction. (3) We have changed the table 2 according to the suggestion in Q8.
> > >
> > > 7. Q7: Please refer to our general reply of baseline with hybrid training. It shows that the LLM tends to perform random reasoning action selection after hybrid training rather than select the best action automatically, demonstrating the necessity of our searching steps.

---

> > > > ### Comment · Reviewer_Npg3 · 2024-11-26
> > > >
> > > > Thanks for your response. It solved most of my concerns, I have raise my rating.

---

> > > > > ### Author Response · Authors · 2024-11-27
> > > > >
> > > > > We sincerely thank the reviewer for raising the score and providing detailed comments and insights that have greatly contributed to a deep analysis of our paper. We have added the discussion in the revised version of the paper.

---

### Official Review · Reviewer_6PPW · 2024-10-25

**Soundness:** 3
**Presentation:** 2
**Contribution:** 2
**Rating:** 6
**Confidence:** 4

**Summary:**

The paper proposes DOTS, which can dynamically choose prompting methods to do reasoning, by dividing reasoning into analysis, solution, and verification modules, and choosing from recent popular methods for each module.

The dynamic planner can be either external or internal to incorporate various LLMs with difference sizes and openness.

Experiments are conducted on in-distribution, near-distribution, and out-of-distribution settings, and zero-shot and few-shot evaluations, where DOTS outperforms most recent LLM prompting reasoning methods.

**Strengths:**

DOTS unifies many of recent LLM prompting reasoning methods, and can dynamically choose better method for each module for each data sample. It borrows the strength of various solutions in literature, and can possibly incorporate future works.

The paper contains comprehensive experimental results, on various datasets across domain, and in/near/out-of-distribution experiments. The experiments show superior performance of DOT to previous prompting methods consistently.

The paper also includes ablation studies to verify the importance of each of the breakdown modules.

It also includes analysis of preferences of DOTS on different tasks, with insightful explanations. It also includes an analysis of efficiency, showing comparable computing costs to other advanced prompting techniques.

**Weaknesses:**

Although DOTS has higher average performance, DOTS cannot consistently beat baselines in all tasks. This is a little surprising as DOTS should be able to be considered as superset of all baselines. On those dataset DOTS fall behind baselines, it would mean it's not necessary to do dynamical reasoning (as one can choose the baseline instead of choosing different modules for each data sample).

This phenomenon is more often in out-of-distribution settings. The average score on out-of-distribution also shows smaller margin. This means lack of training (in a domain) can lead to much worse performance of DOTS. However, as also mentioned by author, such training set may be not available, casting generalizability concern on DOTS.

It would be beneficial to show the comparison on near-distribution zero-shot settings (e.g. the few-shot datasets in the paper) to verify that only near-distribution training is needed, but the paper only includes few-shot settings on these datasets. (also see questions)

To clarify, the few-shot experiments with consistent outperformance can partly address the point, but I would expect better zero-shot performance since training is involved.

**Questions:**

+ (Type) Equation 2: $T$ has not been defined.
+ (Clarification) For the "few-shot" settings, do you use the planner trained on MATH dataset? If so, they should be "near-distribution + few-shot", since those "few-shot" datasets are math-related.
+ Since DOTS has marginal performance on OOD tasks (despite DOTS need additional training), what's the performance on near-distribution (e.g. the few-shot datasets in the paper) but zero-shot settings?
+ In Table 3, 4, 5, on a few datasets DOTS is not the best among methods. Does it mean in some datasets, it's better to choose one single baseline method instead of choosing modules dynamically as in DOTS?
+ DOTS select one reasoning method (one choice for each module). Would it be better to attempt multiple choices and choose the best one (and this would align more with "trajectory search")?
+ (Minor) It would be beneficial to also include the computing efficiency of external DOTS.

---

> ### Author Response · Authors · 2024-11-25
> **Reply for reviewer 6PPW**
>
> Thank you very much for your insightful comment.
>
> 1. W1 and Q4: We thank the reviewer for this insightful question. For certain specific datasets, such as Game of 24, there exists a single reasoning path (PoT and verification) that consistently delivers the best performance. Therefore, the upper boundary for our method for this Game of 24 is PoT and verification (i.e., “self-refine” in our tables). Our method achieves comparable results to this upper boundary and outperforms other methods greatly, demonstrating our method can identify and select the best reasoning path for such problems, which is a satisfying result. Besides, in practical scenarios, questions from the user are not always in the same pattern and we cannot control the types of questions users may ask. Therefore, we argue that the average accuracy and the accuracy on datasets such as MATH, which encompass diverse questions, is a more significant indicator of our method’s utility. From our results, we see that dynamic reasoning, where an LLM flexibly adapts its reasoning trajectories, is absolutely necessary for the model to perform consistently and stably well across tasks. We kindly refer the reviewer to our General Response for a clarification of our goal in this work.
>
> 2. W2: For OOD results, please refer to our general reply.
>
> 3. W3 and Q3: We sincerely thank the reviewer for providing constructive feedback. To demonstrate near-distribution zero-shot settings, we selected two datasets from the big bench but not in BBH: auto-categorization and last-letter concatenation datasets. Both are reasoning datasets and represent near-distribution and zero-shot scenarios. We tested it with the Llama-3-8b-Instruct as the task-solving LLM.
>
> | Method             | Auto Categorization | Last Letter Concatenation |
> |--------------------|---------------------|---------------------------|
> | CoT               | 50.7               | 4.7                       |
> | LTM               | 46.5               | 3.3                       |
> | PA                | 45.3               | 6.5                       |
> | PoT               | 37.2               | 90.0                      |
> | Self-refine       | 37.4               | 85.9                      |
> | Vanilla (CoT) SFT | 49.7               | 7.6                       |
> | Hybrid SFT        | 39.2               | 81.3                      |
> | DOTS              | 51.6               | 89.3                      |
>
> From this table, we can find that our method performs better than prompting engineering and SFT methods in near-distribution zero-shot scenarios. It can select the appropriate action for each task (CoT for the auto categorization dataset and PoT for the last letter concatenation).
>
> 4. Q1: We apologize for the confusion and have revised the pdf.
>
> 5. Q2: Thank you for your comments. We trained the model using the few-shot datasets and the MATH dataset together. Regarding whether these few-shot datasets can be considered "near-distribution" to the MATH dataset, we believe this concept is somewhat vague, as mathematics and the few-shot datasets, such as symbolic reasoning and games, are not entirely related. However, we will revise the experiment section to provide a clearer statement of our data used in training.
>
> 6. Q3 and Q4 are answered above.
>
> 7. Q5: The idea of trying multiple reasoning attempts during inference and selecting the best result is very interesting. Our method can be seen as a single-shot planning and the mentioned idea can be seen as an exploration and selection process. The main reasons we use the single-shot manner are: 1) it’s faster and computationally efficient; and 2) selecting from multiple generated plans relies on having a reliable verifier, which is another open problem to explore.
>
> 8. Q6: For external planning methods, since our planning LLM is only an 8B model and the generation output is just a plan and not long (the number of tokens ranging from 100~200), the additional computing is very limited compared to the computing cost of the solving step with larger task-solving LLMs (such as 70B models or GPT-4).

---

> > ### Comment · Reviewer_6PPW · 2024-11-28
> >
> > Thanks for the author response. It addresses much of my concerns. I would like to keep my score, which is already a positive rating.

---

### Official Review · Reviewer_in1A · 2024-11-01

**Soundness:** 3
**Presentation:** 3
**Contribution:** 3
**Rating:** 5
**Confidence:** 3

**Summary:**

The paper proposes a new way to learn reasoning through two learning paradigms. The paper (DOTS) proposes both a way to FT an planner LLM and in directly FT-ing the task solver.

**Strengths:**

1. Adaptability of the method: The method can be used either on the planner or on the task solver.

2. The method shows lower cost than non-CoT methods.

3. Improved Reasoning ability: The paper demonstrates improved reasoning ability by allowing for dynamic selection for a given question, outperforming static methods as well as self refinement (most but not all scenarios).

**Weaknesses:**

1. The proposed method is complex and involves several steps. It is not clear if the complexity is warranted and ablation studies on the various aspects can help.

2. The paper does not explore decomposition or tool use which would be crucial for complex tasks.

3. The paper glosses over how the atomic trajectories are collected.

**Questions:**

Q1. How many trajectories are required to improve performance substantially at FT-time? A more thorough discussion would be useful.

Q2. How does the proposed approach scale to more complex problems where process rewards would be beneficial?

---

> ### Author Response · Authors · 2024-11-25
> **Reply to the reviewer in1A**
>
> We thank the reviewer for the thoughtful comments and hope to provide clear explanations during the discussion.
>
> 1. W1: We need to point out that our method is not complex in both the training and inference phases. During the training phase, we: (1) search for the optimal trajectory; (2) ask GPT-4o to analyze why this trajectory is optimal and add explanations; and (3) train the LLM to predict the optimal reasoning trajectory before solving the problem. In the inference phase, we only need to sample the plan onceone time. We have included an ablation study in section 3.6 to highlight the importance of the search process and the inclusion of explanations. It shows the effectiveness of each step designed in our method.  The consistently strong performance of DOTS on various datasets and tasks also justified our overall approach design.
>
> 2. W2: We kindly note that our design includes decomposition (shown in line 177, Analysis Layer). Using a Python Interpreter as a tool is a common strategy when solving reasoning problems, and our exploration also incorporates this aspect (see line 181, Solution Layer). The main idea of the current version is to show the effectiveness of our idea. Adding more tools is naturally easy in our design as we could just add more possible actions in the Solution layer.
>
> 3. W3: We would like to clarify that the term "atomic trajectory" was not explicitly used in our paper. If you are referring to "atomic actions", these are based on prior work, as detailed in the first paragraph of Section 2.2. Alternatively, if your question refers to the "action trajectory after searching", the corresponding process is outlined in Section 2.3 and Algorithm 1. Specifically, this involves first enumerating all possible action trajectories, followed by testing each trajectory and collecting those that achieve higher accuracy. We hope this explanation addresses your query. If there is any additional information you would like us to elaborate on this question, we would be more than happy to engage in a deeper discussion.
>
> 4. Q1: Our few-shot setting demonstrates that our method can achieve improvements even with a limited number of instances and trajectories. However, we argue that few-shot examples help the LLM grasp high-level action planning ideas, such as "if the question involves math, use PoT." In contrast, more extensive training instances are necessary to teach the LLM fine-grained question-specific strategies, such as "use PoT when dealing with counting", "use CoT for geometric reasoning.", and "use verification for solving equations". Therefore, we argue that more training instances and trajectories would make planning ability better.
>
> 5. Q2: We appreciate the reviewer for raising this important future exploration. To extend this work with a process reward, we would like to mention a possible future work for searching for appropriate action in each process step: Assuming we have access to a process rewards model, our method could be applied stepwise by searching for the optimal action in each step. For example, the search might reveal that using a Python built-in function and verifying the result in step 1 achieves a higher process reward while applying query decomposition and CoT reasoning in step 2 yields better results. We could then construct training data with this trajectory, enabling it to learn the most appropriate action for each step.

---

> ### Author Response · Authors · 2024-12-02
> **A kind reminder of the end of the discussion period (December 2nd)**
>
> Dear Reviewer in1A,
>
> As we are approaching the end of the discussion period, we want to kindly remind you to read our response and share any further concerns you have. We are happy to address them before the end of the discussion. Thank you!

---

### Official Review · Reviewer_kuiZ · 2024-11-03

**Soundness:** 3
**Presentation:** 3
**Contribution:** 3
**Rating:** 8
**Confidence:** 4

**Summary:**

This paper proposes DOTS, an approach enabling LLMs to reason dynamically via optimal reasoning trajectories search, tailored to the specific characteristics of each question. DOTS first defines atomic reasoning action modules, then searches for the optimal action trajectory for each training question through iterative exploration and evaluation for the specific task-solving LLM, and uses the collected trajectories to train a model to plan for the reasoning trajectories of unseen questions.

**Strengths:**

1. The authors propose a dynamic reasoning method which can enable the model to decide the appropriate atomic actions based on the characteristics of the input question.
2. The authors conduct comprehensive experiments to prove the effectiveness of the proposed method, containing in distribution, few-shot, and out-of-distribution settings.
3. The proposed method can be used on both open-source and close-source models.

**Weaknesses:**

1. The method proposed in this paper does not show significant improvement on out-of-distribution (OOD) tasks, and it incurs additional computational overhead compared to prompt engineering methods.
2. The baseline for Vanilla SFT only used the training data from CoT. I believe it should be compared with baselines from other reasoning formats to demonstrate the effectiveness of the proposed method, such as using the program reasoning format and mixed training data from CoT and Program.
3. The experiments demonstrating the effectiveness of the Search section are compared with randomly selected reasoning paths. I am curious about how the results would compare if we contrasted it with reasoning paths generated by combining non-empty actions from each layer.
4. Using PoT prompts on non-code models may yield suboptimal results. Would combining PoT with code models improve the results of the PoT baseline?

**Questions:**

See weaknesses.

---

> ### Author Response · Authors · 2024-11-25
> **Reply for reviewer kuiZ**
>
> Thank you for your valuable comments. We have tried addressing each of the concerns below:
>
> 1. W1: For the concern about the OOD performance, please refer to our general reply. Besides, regarding additional computational costs compared with the prompting engineering method, we need to emphasize that our method does not introduce much additional computational cost during the inference stage. For external planning methods, since our planning LLM is an 8B model, the additional cost is very limited compared to larger task-solving LLMs (such as 70B models or GPT-4). For internal planning methods, in section 3.8, we compared our approach with prompt engineering methods, demonstrating that our method won’t cause additional computational overhead.
>
> 2. W2: Thanks for the suggestion. For this discussion, please refer to our general reply for the baselines of the hybrid training.
>
> 3. W3: For showing the combining of all non-empty actions, we added an extra baseline for using all three layers with non-empty action selection. Please refer to our general reply. Our method still outperforms this baseline.
>
> 4. W4: We sincerely appreciate the reviewer's thoughts; it is very helpful. However, it is out of the initial scope of our paper. This is because (1) We focus on the general and various reasoning tasks, not a specific task. Therefore, a code-based model, which may not be suitable for answering text-heavily reasoning tasks, is unlikely to be a candidate for the task-solving LLM compared with the general textual generation LLM. (2) If we consider a scenario where the task is only for code generation and the task-solving LLM is a code-based LLM, it will result in more successful PoT paths in searching. Consequently, our search result will always be the PoT and the tuned planner would tend to select PoT in all situations, which gives the exact same performance of using PoT.

---

> > ### Comment · Reviewer_kuiZ · 2024-11-26
> >
> > Thanks for your response.
> > The additional experimental results are solid.
> > I think it is a good paper and give me some inspirations. Therefore I raise my rating.
> > Good luck!

---

> > > ### Author Response · Authors · 2024-11-27
> > >
> > > Thank you for your recognition of our work! We deeply appreciate the reviewer’s efforts in offering insightful feedback on OOD discussion and more baselines, which helped us significantly improve the paper's strengths.

---

### Author Response · Authors · 2024-11-25
**General Reply**

We appreciate all the reviewers for their valuable feedback and constructive suggestions. In this message, we summarized some common concerns and our response.
1. Performance of OOD tasks:

Before responding to the shared concern about our OOD result, we would like to clarify the goal of this work. Our goal is to have a planning strategy that universally works well for all kinds of reasoning problems, and the scenario we target is that in an open-world setting, users of an LLM can ask any questions. In other words, we are not building any task-specific planner and do not know the question type before the user asks a question. Therefore, picking a pre-defined reasoning trajectory optimized for a single task intuitively would not be an optimal strategy. Instead, we aim to devise a dynamic reasoning approach that has the flexibility to plan for the optimal reasoning trajectory based on the user question or task at the test time. Under this scope of work, we are more concerned about whether a dynamic reasoning approach can achieve strong performance _consistently across tasks_, rather than beating the state-of-the-art approach on any single task, as we did not introduce new reasoning actions.

As for our OOD experimental results, Reviewer 6PPW expressed the concern that DOTS did not outperform the best approaches on all tasks. As our method only plans on the use of existing modules but does not introduce any new reasoning components, we do not expect DOTS to substantially outperform the best pre-defined reasoning path in any single dataset, especially for datasets where the type of questions is not diverse and one single trajectory is much better than all others. In our target scenario, an ideal planner should (1) achieve performance that surpasses any single static trajectory on datasets that require diverse optimal trajectories, such as MATH, and (2) maintain consistently on-par performance with the best static prompting method on datasets where a single trajectory is always optimal, such as Game of 24. Our experiments confirm that our method achieves this goal better than any baseline, demonstrating that DOTS can make the LLM more adaptable to different tasks.

The second common concern is that the improvement of DOTS is not significant compared to baselines. As discussed earlier, our method does not introduce additional components or actions, so significant improvements should not be expected across every single task. However, our approach always obtains a comparable or best result consistently across all tasks. The strongest baseline to us is “self-refine”, which integrates PoT and verification. Because for two out of the four OOD datasets (i.e., DeepMind-Math and MMLU-pro), PoT and verification are mostly sufficient, DOTS’s performance is either comparable or slightly better (in the latter case, it is often a result of additionally applying an analysis component). On the other two datasets (i.e., StrategyQA and DROP), PoT is not a preferable choice, yet DOTS flexibly selects CoT as a better action; as a result, the strong baseline “self-refine” performs 2-3% absolute accuracy lower than DOTS. Overall, DOTS consistently performs better or on par with “self-refine”, which shows the strong advantage of dynamic reasoning.

---

> ### Author Response · Authors · 2024-11-25
> **General Reply 2**
>
> 2. More baselines
>
> We added two sets of baselines in Appendix E. Our method still achieves significant improvement compared with these two baselines.
> The first set of baselines is about using all three layers in a static manner with the non-empty action selection. We reported the results when pairing different choices across the three layers (i.e., {Rewrite, Decompose} x {CoT, PoT} x {Verification}, a total of 4 baselines for each solver LLM setting). The result shows that our method still outperforms using all three layers. For detailed results, please refer to Appendix E in pdf.
>
>
> The second baseline is about hybrid training, i.e., using a mixture of training data with both CoT and PoT. We use the same training instance and each instance is provided with both the PoT and CoT reasoning paths. We reported the results in Appendix E. The result shows that our method still outperforms the hybrid training. From the results, it can also be observed that for datasets where CoT outperforms PoT, training solely with CoT yields better performance. We analyzed several cases and found that although mixed training allows the LLM to choose between CoT and PoT dynamically, the results tend to perform random selection rather than actively analyzing the problem and selecting the most suitable path for the given question carefully because we didn’t guide the LLM learn this. This highlights the importance of searching for the optimal reasoning path before tuning the LLM.
>
>
> | Method             | MATH | BBH  | Game of 24 | TheoremQA | Deepmind-math | MML U-pro | Strategy QA | DROP | Average |
> |--------------------|------|------|------------|-----------|---------------|-----------|-------------|------|---------|
> | Vanilla (CoT) SFT | 33.9 | 61.0 | 18.5       | 14.8      | 39.6          | 40.3      | 71.8        | 49.0 | 36.7    |
> | Hybrid SFT        | 32.3 | 53.3 | 18.7       | 15.6      | 49.1          | 36.5      | 65.9        | 46.2 | 39.7    |
> | DOTS              | 34.4 | 69.7 | 21.9       | 16.1      | 55.3          | 39.7      | 68.2        | 48.8 | 44.3    |

---

### Meta-Review · Area_Chair_msCR · 2024-12-24

**Metareview:**

This paper proposes to enhance LLM reasoning via dynamically tailoring trajectory search. The proposed method should be generally applicable. The experiments show that its inference cost could potentially be lower than non-CoT methods. The proposed method is intuitive. However, the experiments, especially the setting could be more principled. Some of the comparison may not be completely fair to the baselines when considering the details. Its OOD improvement seems limited, considering it requires SFT before deploying the method. Thanks to some of the added new experiments, some of the concerns is alleviated.

**Additional Comments On Reviewer Discussion:**

The paper is easy to follow. The proposed method is intuitive to understand. Reviewers were balancing between its improvement and its additional cost. Furthermore, whether the experiments (more importantly, their settings) are principled enough. The rebuttal alleviates some of the concerns regarding the experiments. And hence overall, it leans towards acceptance.

---

### Decision · Program_Chairs · 2025-01-22

Accept (Poster)